# The dynamics of γδ T cell responses in nonhuman primates during SARS-CoV-2 infection

Alyssa C. Fears[1], Edith M. Walker[1], Nicole Chirichella[1], Nadia Slisarenko[1], Kristen M. Merino[1], Nadia Golden[1], Breanna Picou[2], Skye Spencer[2], Kasi E. Russell-Lodrigue [3], Lara A. Doyle-Meyers [3], Robert V. Blair [4], Brandon J. Beddingfield [1], Nicholas J. Maness [1,5], Chad J. Roy [1,5] & Namita Rout [1,5,6 ✉]

Although most SARS-CoV-2 infections are mild, some patients develop systemic inflammation and progress to acute respiratory distress syndrome (ARDS). However, the cellular mechanisms underlying this spectrum of disease remain unclear. γδT cells are T lymphocyte subsets that have key roles in systemic and mucosal immune responses during infection and inflammation. Here we show that peripheral γδT cells are rapidly activated following aerosol or intra-tracheal/intra-nasal (IT/IN) SARS-CoV-2 infection in nonhuman primates. Our results demonstrate a rapid expansion of Vδ1 γδT cells at day1 that correlate significantly with lung viral loads during the first week of infection. Furthermore, increase in levels of CCR6 and Granzyme B expression in Vδ1 T cells during viral clearance imply a role in innate-like epithelial barrier-protective and cytotoxic functions. Importantly, the early activation and mobilization of circulating HLA-DR$^+$CXCR3$^+$ γδT cells along with significant correlations of Vδ1 T cells with IL-1Ra and SCF levels in bronchoalveolar lavage suggest a novel role for Vδ1 T cells in regulating lung inflammation during aerosol SARS-CoV-2 infection. A deeper understanding of the immunoregulatory functions of MHC-unrestricted Vδ1 T cells in lungs during early SARS-CoV-2 infection is particularly important in the wake of emerging new variants with increased transmissibility and immune evasion potential.

[1] Division of Microbiology, Tulane National Primate Research Center, Covington, LA, USA. [2] High Containment Research Performance Core, Tulane National Primate Research Center, Covington, LA, USA. [3] Division of Veterinary Medicine, Tulane National Primate Research Center, Covington, LA, USA. [4] Division of Comparative Pathology, Tulane National Primate Research Center, Covington, LA, USA. [5] Department of Microbiology and Immunology, Tulane School of Medicine, New Orleans, LA, USA. [6] Tulane Center for Aging, Tulane University School of Medicine, New Orleans, LA, USA. ✉email: nrout@tulane.edu

The coronavirus disease 2019 (COVID-19) pandemic caused by severe acute respiratory syndrome (SARS) coronavirus-2 (SARS-CoV-2) has affected more than 273 million people worldwide and resulted in more than 5 million deaths as of December 2021[1]. SARS-CoV-2 exposure may result in a clinical spectrum ranging from asymptomatic/very mild disease to life-threatening acute respiratory distress syndrome (ARDS)[2–4]. With emerging new variants, there is an urgent need to understand the immune mechanisms of protection from severe respiratory distress during SARS-CoV-2 infection. Studies have revealed that the ARDS due to severe COVID-19 is associated with acute lymphopenia[5]; excessive production of proinflammatory cytokines, including IL-1β, IL-6, and TNF-α[6] and activation of B and T cells, NK cells, innate lymphoid cells, and myeloid cells including neutrophils and monocytes[7]. However, very little is known about the role of γδT cells, a key innate-like immune cell type capable of early virus detection and rapid initiation of antiviral functions by direct cytotoxicity, cytokine secretion, recruitment, and activation of other innate and adaptive immune cells.

γδT cells represent 1–5% of blood lymphocytes and a much larger fraction of T cells in mucosal tissues, such as the lung and gut[8]. Primate γδT cells are generally divided into two major populations based on the Vδ TCR chain usage: Vδ1 and Vδ2 subsets[9]. γδT cells have been implicated in recovery from influenza infection[10] and are required for the initiation of effective immune responses and maintenance of lung homeostasis in influenza-infected neonates[11]. A study of healthcare workers who survived SARS-CoV infection during the 2003 epidemic, revealed selective in vivo expansion of effector memory Vδ2 T cells 3 months after the onset of disease[12]. Furthermore, this expansion was associated with higher anti-CoV IgG titers, and the Vδ2 T cells displayed an IFN-γ-dependent ability to directly kill CoV-infected target cells in vitro, suggesting their protective role in SARS infection[12]. In the context of COVID-19, an early comprehensive analysis of the myeloid and lymphoid immune cell subsets in a cohort of patients with clinical manifestations ranging from mild to fatal outcomes revealed that the ratio of immature neutrophils to Vδ2 T cells strongly predicted the onset of pneumonia and hypoxia[13], indicating that higher ratios with lower γδT cells are associated with severe outcomes. Moreover, lower circulating frequencies of γδT cells[14–16], specifically the Vδ2+ subsets were reported in hospitalized patients, along with a highly preferential shift in effector memory phenotype during severe disease in patients that succumbed to SARS-CoV-2 infection[17,18]. While these reports in patients with severe COVID-19 disease indicate that activated γδT cells are among the cells decreased during lymphopenia and ARDS, it is unclear whether their decline is due to increased migration to the lungs or due to activation-induced cell death (AICD). Moreover, how their precise functions and associated mechanisms during early infection may predict disease outcome remains to be elucidated. This knowledge would be essential for the development of vaccines aimed to stimulate both innate and adaptive immunity against emerging variants.

In this study, we investigated the dynamics of γδT cell frequencies and functions through the course of SARS-CoV-2 infection in two nonhuman primate (NHP) species, including rhesus macaques (RM) and African green monkeys (AGM), that were inoculated via the aerosol exposure or intra-nasal/intra-tracheal (IN/IT) deposition routes to determine the impacts of route of SARS-CoV-2 infection on COVID-19 disease characteristics. Our results indicate that Vδ1+ γδT cells are involved in the early immune response to SARS-CoV-2 infection in NHPs and the mode of virus deposition in the airways influences the quality of this early immune response.

## Results

### Experimental design for two different routes of SARS-CoV-2 infection and blood cell ratios during the course of infection.
To explore the kinetics of early immune responses to SARS-CoV-2 infection, we leveraged a study examining two potential human exposure routes simulated by utilizing small-particle aerosol or combined intratracheal and intranasal (IT/IN) deposition in 16 NHPs comprising of eight adult AGM and RM (Fig. 1). The viral dynamics and clinical pathology are described in detail elsewhere[19]. Briefly, viral loads reached their peak between 1-7 days postinfection (dpi), with BAL registering higher peak at 1dpi and faster decline by 7dpi among all the mucosal sites (Supplementary Fig. 1). Despite high viral loads in mucosal compartments ($10^7$-$10^9$ copies per swab or mL), only mild clinical signs were observed compatible with previous reports of resolving SARS-CoV-2 infection in NHPs[20–23]. These included mild dyspnea and malaise (refusal to move around the cage), which were observed more readily in the IT/IN-exposed NHPs (Supplementary Fig. 2).

### Aerosol SARS-CoV-2 infection leads to rapid activation and expansion of circulating γδ T cells.
γδ T cells are highly conserved among humans and NHPs, particularly the gene segments of the δ locus that have not changed substantially in the primate

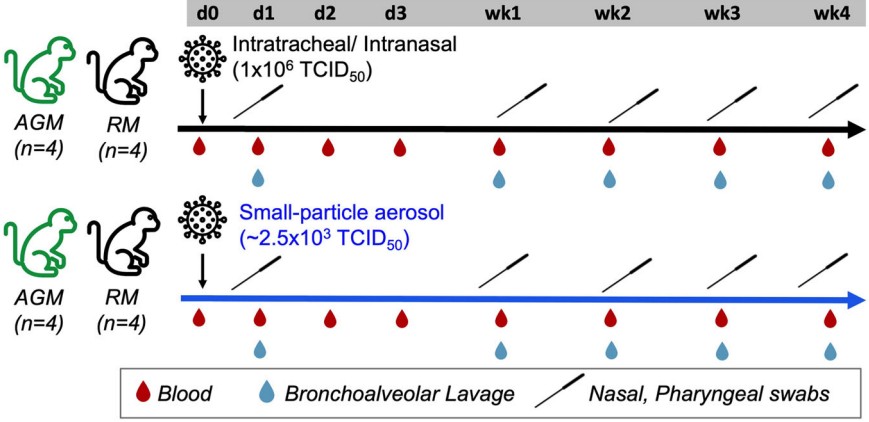

**Fig. 1 Study Design.** Rhesus macaques (RM, $n = 8$) and African green monkeys (AGM, $n = 8$) were split into 4/group and inoculated with SARS-CoV-2 via either intratracheal/intranasal (IT/IN) route or aerosol route Serial collections of blood, bronchoalveolar lavage, and swabs of the nasal and pharyngeal cavities were taken at several study timepoints, including d0, d1, d2, d3, wk1, wk2, wk3, and wk4 post-infection.

lineage from humans to marmosets[24–26]. Thus, we combined data from the two NHP species in our analyses focused on kinetics of γδ T cell responses to SARS-CoV-2 infection. As anticipated, no differences were observed in the baseline γδ T cell characteristics between AGMs and RMs (Fig. 2b). Evaluation of early events in circulating T lymphocyte subsets, including mucosal-associated invariant T (MAIT) cells, CD4 T cells and CD8 T cells (Fig. 2a) revealed distinct responses between aerosol and IT/IN routes of exposure. A significant increase in γδ T cell frequencies and absolute counts was observed in the aerosol-exposed AGMs and RMs at 1-2dpi, which returned to baseline by 3dpi (Fig. 2b, d). Concomitant to increased γδ T cell numbers, a significant decline in CD4 T cell frequencies and counts was noted in the aerosol group on 1dpi (Fig. 2b, d). In contrast, the IT/IN animals showed a trend of delayed increase in γδ T cell counts at 7dpi, and a significant increase in CD8 T cell counts at 14dpi (Fig. 2c, e). The early expansion of circulating γδ T cells at 1dpi in aerosol exposure was accompanied by increased HLA-DR expression (Fig. 3a). Of note, there was a delayed increase in activated γδ T cells at 3dpi in the IT/IN group as well (Fig. 3b), albeit without a significant increase in circulating numbers. MAIT cells also showed a similar trend with early activation in the aerosol group and delayed response in the IT/IN group (Fig. 3a, b). This suggests that nonclassical T cells, particularly γδ T cells, respond rapidly to aerosol SARS-CoV-2 infection and the kinetics of this response is divergent between the two routes of exposure and consistent between AGM and RM. This was also evident in the decline of activated CD8 T cells at 7dpi in the aerosol group in contrast to the significant increase between 3-7dpi in the IT/IN group (Fig. 3a, b). Notably, γδ T cell frequencies strongly correlated with 1-week viral loads in the BAL following aerosol exposure and showed a trend of positive correlation in the IT/IN inoculated group (Fig. 3c), suggesting their expansion in response to ongoing virus replication in the lungs. Subsequently, concordant with peripheral blood γδ T cell decline in COVID patients[17], γδ T cell frequencies declined significantly in both aerosol and IT/IN groups (Fig. 2b, c) during virus clearance from the lower respiratory tract around 28dpi, suggesting likely recruitment to sites of inflammation and tissue repair following viral clearance.

**Enhanced cytotoxic and Th17-type functional phenotype of circulating γδ T cells during SARS-CoV-2 infection.** Based on the significant activation of circulating γδ T cells and correlation with early viral loads, we next examined the dynamic changes in their functional phenotype during active viral replication in the first two weeks of infection. Expression of CXCR3, CCR6, and Granzyme B (GrB) as markers of Th1, Th17, and cytotoxic functionality was evaluated in peripheral blood γδ T cells at baseline and 3, 7, and 14dpi (gating strategy in Supplementary Fig. 3). CXCR3-expressing T cells are important in antiviral immune responses[27,28] and reported to be higher in the blood of severe COVID-19 patients[29]. Accordingly, we observed an early and transient increase in CXCR3 expression on circulating γδ T cells at 1dpi following aerosol SARS-CoV-2 exposure in the NHPs (p = 0.03; Fig. 4a). In contrast, the IT/IN exposed animals did not show any significant changes in frequencies of CXCR3+ γδ T cells until at least 3dpi and displayed a significant decline in CXCR3 expression later at 7dpi that persisted through the 4-week time-point (p ≤ 0.006; Fig. 4a), suggesting likely recruitment to lungs and other sites of viral replication. However, CCR6 expression, the chemokine receptor that enables homing of Th17-type cells to the lung and the gut[30], increased significantly on γδ T cells in both routes of exposure indicating a skewing toward IL-17-producing functionality. This increase was evident earlier in

the IT/IN group (between 3-7dpi) in contrast to 7-14dpi in the aerosol group (Fig. 4a), suggesting an early shift from Th1-type functionality toward epithelial barrier protective potential in the setting of clinical symptoms in contrast to the delayed increase in the setting of milder disease symptoms. This differential expression of CXCR3 and CCR6 was not consistently observed in other T cell subsets, including CD4 T cells, CD8 T cells, and MAIT cells (Supplementary Fig. 4).

Besides cytokine production, γδ T cells exert their cytotoxicity on virus-infected cells by releasing perforin and granzymes that are contained in their lytic granules[31–33]. Significantly greater frequencies of GrB and perforin expressing CD4 and CD8 T cells have been reported in severe COVID-19 cases than in the mild group during the later phase of illness[34]. Quantification of intracellular GrB expression in γδ T cells in our study revealed a significant increase at 7dpi in both routes of exposure that was maintained at 14dpi in the IT/IN group (Fig. 4a). To gain more insight into dynamic changes in functional phenotype during ongoing virus replication, we next assessed the γδ T cell polyfunctional phenotype with regard to combinatorial expression of CXCR3, CCR6, and GrB during the first two weeks of SARS-CoV-2 infection. At baseline, circulating γδ T cells displayed significant Th1-type functional phenotype with dominant CXCR3 expression and moderate levels of GrB (Fig. 4b), with a small proportion co-expressing CXCR3/CCR6 conforming to their shared Th1/Th17 functionality in primates[35]. Following SARS-CoV-2 exposure, a highly significant early increase by 3 dpi in activated CCR6+ γδ T cells was observed in the IT/IN exposure group comprising of monofunctional CCR6+ cells and CXCR3-coexpressing polyfunctional cells (Fig. 4b, Supplementary Fig. 5), suggesting enrichment of Th17-type effector functionality. The circulating γδ T cells in the aerosol group, however, did not display increased CCR6 expression until 7dpi (Fig. 4b, Supplementary Fig. 6). This was consistent with the later onset of signs of SARS-CoV-2-related disease observed in this group (Supplementary Fig. 2). Notably, the GrB+ γδ T cells at 7dpi in IT/IN group comprised CCR6- and CXCR3- cells (Fig. 4b) suggesting a role for exclusively cytotoxic γδ T cells in better control of viral loads in the airways at the 2-week time-point. Additionally, γδ T cell polyfunctional phenotype differed most significantly between aerosol and IT/IN exposure at 3dpi (p = 0.0007), which was maintained above baseline at 7dpi (p = 0.017) but was no longer distinct by 2 weeks of infection (Supplementary Fig. 7), suggesting their involvement in very early responses to SARS-CoV-2 infection and the influence of the mode of exposure on their kinetics.

**Vδ1 subpopulation mainly contributes to the early γδ T cell-activation and significant correlation with SARS-CoV-2 viral loads in the lungs.** γδ T cell subsets in human and NHP are divided into two major subpopulations, namely Vδ1 and Vδ2. So far, studies reporting on the relationship of COVID-19 with γδ T cells have mainly focused on the Vδ2 subset[15,17,18,36]. In our comprehensive analyses of both Vδ1 and Vδ2 T cell subsets (Fig. 5a), we found that in concordance with clinical reports[15,18], circulating Vδ2 T cells were significantly reduced at 7dpi in both routes and were consistently reduced through 28dpi in the aerosol group (Fig. 5b). Interestingly, we observed a highly significant and transient increase in circulating Vδ1 T cells in both IT/IN and aerosol groups at 1dpi that returned to baseline by 3dpi (Fig. 5b) and declined significantly by 28dpi. This indicates that although the later decline in total circulating γδ T cell frequencies is accounted for by loss of both Vδ1 and Vδ2 subpopulations, the early activation and expansion of peripheral blood γδ T cells (Figs. 3 and 4) is mainly contributed by responding Vδ1 T cells.

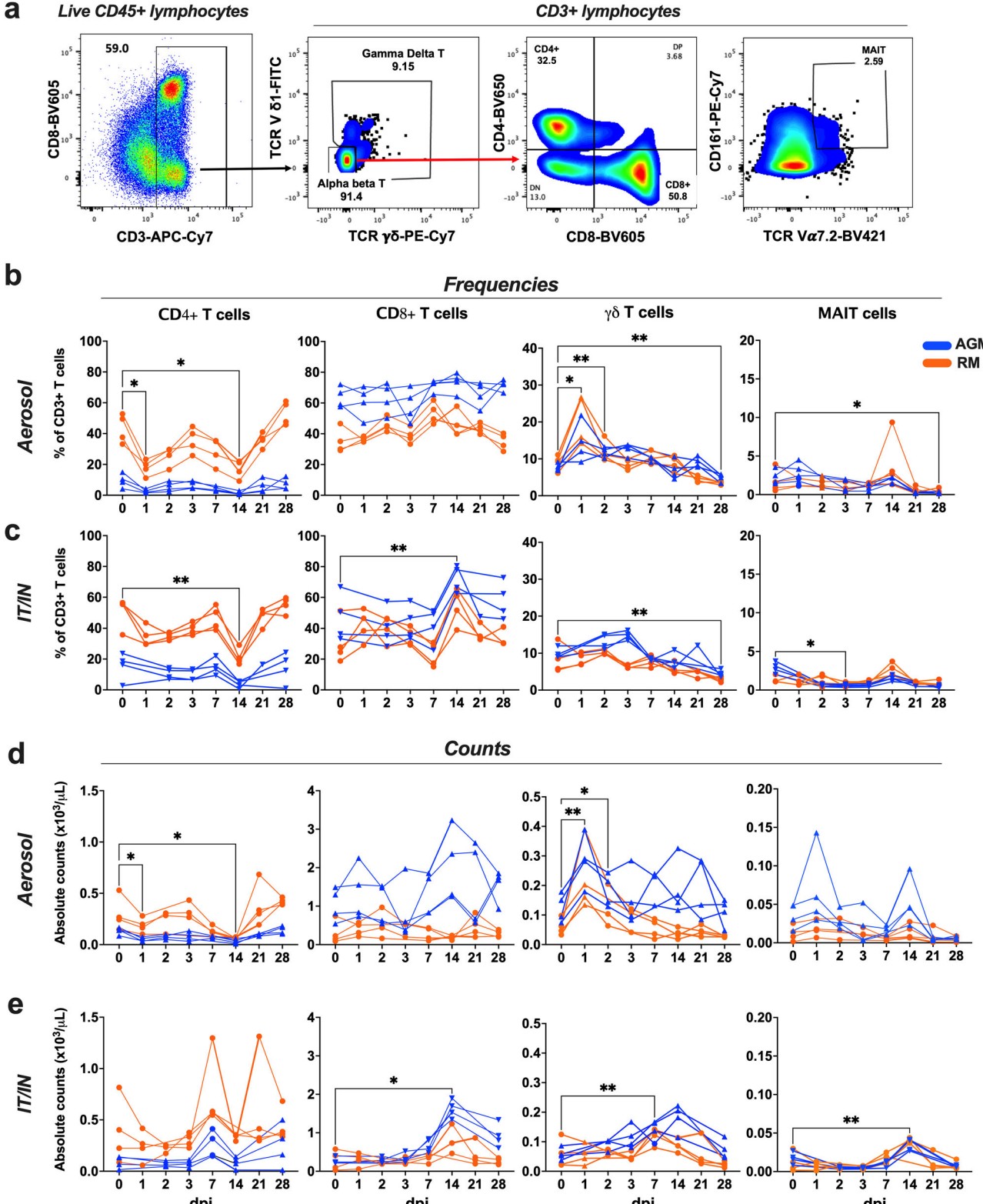

**Fig. 2 Dynamics of circulating adaptive and innate T lymphocyte subsets during SARS-CoV-2 infection via aerosol and IT/IN route in NHPs.**
**a** Representative FACS plots of PBMC showing gating strategy for γδ T, CD4 T, CD8 T and MAIT cells. Quantification of T cell subset frequencies in peripheral blood as percentage of CD3 + T cells following SARS-CoV-2 infection in 8 animals per group either via aerosol (**b**) or IT/IN route **(c)** on 0, 1, 2, 3, 7, 14, 21, and 28 dpi. Absolute counts per μL of blood for γδ T, CD4 T, CD8 T and MAIT cells in aerosol (**d**) and IT/IN (**e**) route of infection. Individual values shown for each animal as symbols connected by lines for RMs (orange) and AGMs (blue). Comparisons of different time points with respect to d0 baseline data were done using repeated measures one-way ANOVA with Dunnett's post hoc tests. Asterisks indicate significant differences between time points (*$p < 0.05$; **$p < 0.01$).

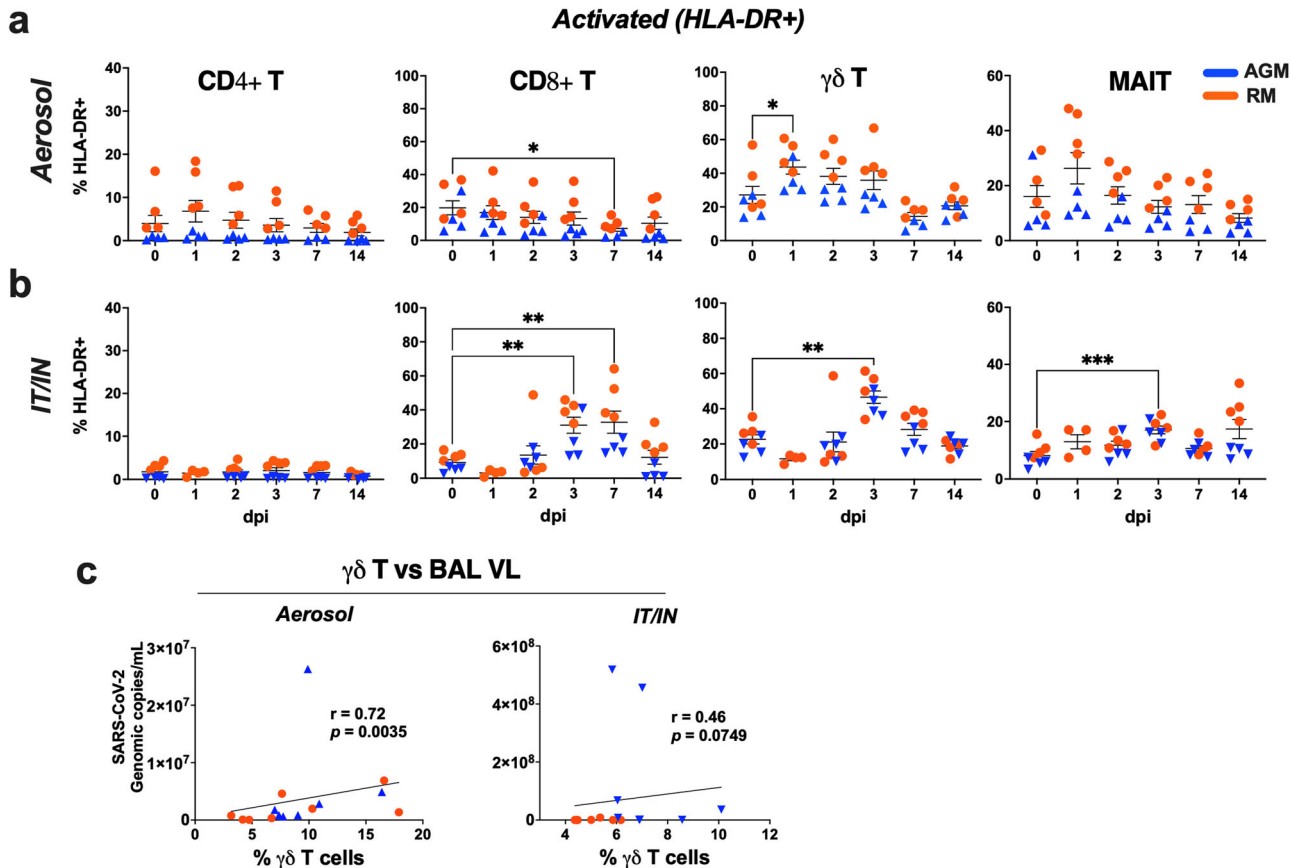

**Fig. 3 Activation kinetics of circulating T lymphocyte subsets and correlation with lung viral loads during early SARS-CoV-2 infection via aerosol and IT/IN route in NHPs.** Frequencies of CD4 T, CD8 T, γδT, and MAIT cells in PBMCs isolated from RM and AGM in aerosol (**a**) or IT/IN group (**b**), showing a rapid increase in activated γδT cells in the aerosol group ($n = 8$/group). Graphs show mean and SEM. Comparisons of different time points with respect to d0 baseline data were done using repeated measures one-way ANOVA with Dunnett's post hoc tests. Asterisks indicate significant differences between time points ($p < 0.05$; **$p < 0.01$). **c** Spearman Rank correlation between BAL viral loads with frequencies of total γδT cells in the first week of infection. Individual values shown for each animal as symbols in orange for RMs and blue for AGMs.

Immunophenotypic analyses of γδ T cells revealed increase in CCR6 and GrB expression and decrease in CXCR3 expression with different kinetics in both routes of exposure (Fig. 5c). The IT/IN group showed an earlier and significant decline in DR+ and CXCR3+ Vδ1 and Vδ2 T cells (Fig. 5c-d) suggesting more rapid recruitment to sites of viral replication and inflammation, which coincided with earlier and more readily detectable clinical symptoms of disease in the animals (Supplementary Fig. 2). Further, Vδ1 T cells displayed a robust and sustained increase in CCR6 and GrB expression, suggesting active polarization toward Th17-type and cytotoxic effector functions following SARS-CoV-2 infection.

Additionally, the frequencies of Vδ1 T cells significantly correlated with BAL viral loads in the first week of infection regardless of the route of exposure (Fig. 5e), indicating that our earlier observation of the overall γδ T cell response to viral burden in the lungs (Fig. 3d) is mainly driven by Vδ1 subsets. The notable changes in the earliest T cell events post-infection at the time of peak viral loads in BAL (1-2dpi following aerosol or IN/IT exposure) with opposing patterns between circulating Vδ1T cells and CD4 T cells were further demonstrated by tSNE plots from flow cytometric analyses (Fig. 6a–b).

**Differential kinetics of activation and induction of functional phenotype of Vδ1 T cells in BAL fluid following aerosol and IT/IN exposure.** To explore the coordination of systemic immune responses in blood with pulmonary responses in our study, we investigated γδ T cell immunophenotype in BAL fluid.

The absence of data from the baseline BAL samples precluded our ability to evaluate changes in γδ T cell frequencies and immunophenotype from pre-infection levels. Thus, we compared the available time-points with healthy control group comprising of uninfected RMs and AGMs, and in the IT/IN group comparisons were made between 14dpi (ongoing viral replication) and 28dpi (nearly resolved infection). Both overall γδ T cell and Vδ1 T cell frequencies were significantly lower in BAL from infected NHPs than healthy controls at 28dpi (Fig. 7a). Vδ1 T cells were highly activated at 14 dpi with significantly higher levels of HLA-DR expression in the IT/IN group in comparison to healthy controls (Fig. 7b). This was attended by higher GrB levels in both Vδ1 and Vδ2 subsets compared to healthy controls (Fig. 7c). Furthermore, GrB expression in the Vδ1 and Vδ2 T cells was significantly higher at 14dpi than at 28dpi (Fig. 7c) consistent with the highest levels of GrB expressing peripheral blood γδ T cells between 7-14dpi, and a gradual decline by 28dpi (Figs. 4a and 5c-d). This suggests that during ongoing viral replication in the lungs, activated γδ T cells increased their cytotoxic potential that later reduced likely due to the release of cytotoxic granules coinciding with viral clearance between 14-28dpi. Interestingly, BAL Vδ1 subsets in the aerosol group had significantly greater intracellular GrB than in the IT/IN group at week-4 ($p = 0.012$), indicating greater cytotoxic potential in the context of a more persistent infection (Fig. 7c). No significant differences in the frequencies of CD4 T, CD8 T and MAIT cells were observed between 14dpi and 28dpi BAL fluid (Supplementary Fig. 8).

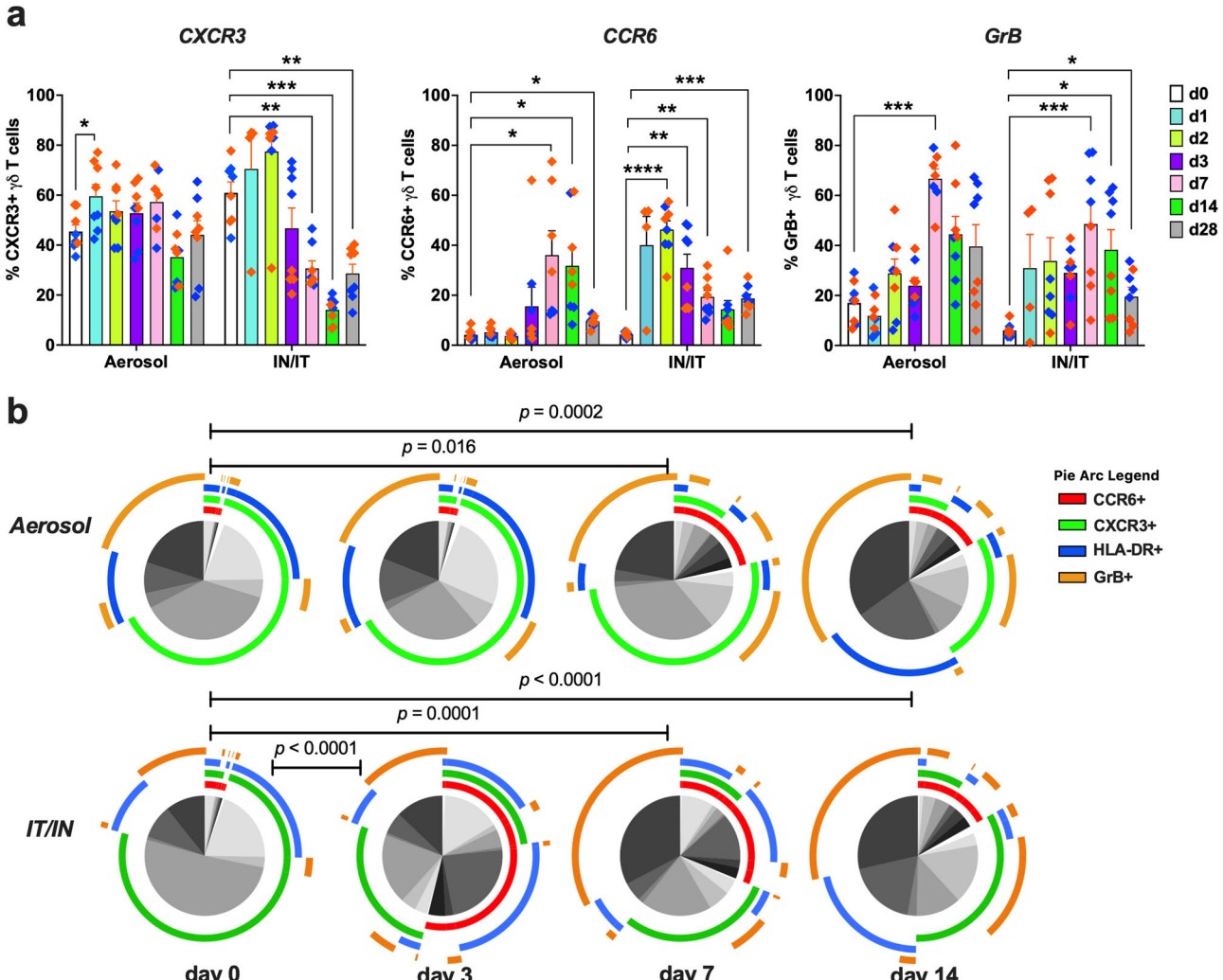

**Fig. 4 Differential increase in circulating γδ T cell GrB and CCR6 expression between aerosol and IT/IN routes of SARS-CoV-2 infection. a** Frequencies of CXCR3 + , CCR6 + , and GrB+ γδ T cells in PBMC of aerosol versus IT/IN exposed RMs (orange diamonds) and AGMs (blue diamonds) through the course of one month SARS-CoV-2 infection (n = 8/group). Graphs show mean and SEM. Comparisons of different time points with respect to d0 baseline data were done using two-way ANOVA with mixed effects model and Dunnett's post hoc tests. Asterisks indicate significant differences between time points (*p < 0.05; **p < 0.01). **b** Pie charts comparing early changes in polyfunctional γδ T cell phenotype from baseline till 2 weeks of infection in the aerosol and IT/IN groups based on the expression of HLA-DR, CXCR3, CCR6, and GrB. The arcs around the circumference indicate each marker expressed by the proportion of cells that lie under the arc. HLA-DR is shown in blue, CCR6 in red, CXCR3 in green, and GrB in orange. p values were computed using the SPICE permutation test.

Additionally, CD161+ Vδ1 T cells were significantly higher at 14dpi than at 28dpi in IT/IN exposed animals (Fig. 7d). We and others have shown that CD161 expression, besides CCR6, is a surrogate marker for Th17-type functions in human and NHP T cells[35,37–39]. Thus, this increase suggests augmentation in epithelial barrier protective function of γδ T cells in the lungs during ongoing viral replication. Overall, despite compartment-specific differences in activation kinetics, both peripheral blood and BAL γδ T cells augmented their cytotoxic and Th17-type functions during ongoing virus replication in the airways.

**γδ T cell functional phenotype parallels BAL cytokine responses during SARS-CoV-2 infection.** In accordance with the mild clinical symptoms in our strudy, serum cytokines/chemokines did not differ significantly between time-points. However, BAL fluid displayed notable differences in the kinetics of cytokine responses between the two routes of exposure. The IT/IN group showed an earlier increase in several pro-inflammatory mediators, including CXCL13, MIP-1α, IL-6, VEGF-D, by 7dpi in contrast to a more delayed increase in the aerosol group at 14dpi (Fig. 8a). On the other hand, MCP-1, IL1RA, sCD40L, FGF-2, and SCF were downmodulated at 7dpi in the IT/IN group and around 14dpi in the aerosol group (Fig. 8a). The earlier increase in local inflammatory mediators and dampening of IL-1RA and SCF in IT/IN exposure and a more delayed increase with per-sistent higher levels of inflammatory cytokines in aerosol group, underscores the differences in the onset of clinical symptoms and kinetics of peripheral T cell responses based on route of exposure.

Notably, the early increase in proinflammatory mediators in BAL fluid following IT/IN exposure (Fig. 8a, right) coincided with a significant activation of circulating γδ T cells and MAIT cells at 3dpi and CD8 T cells at 3-7dpi and (Fig. 3b). In contrast, aerosol exposure during later increase in proinflammatory cytokines at 14dpi (Fig. 8a, left) demonstrated a distinct pattern of immediate increase in activated γδ T cells on 1dpi and decrease in circulating levels of activated CD8 T cells at 7dpi (Fig. 4a). Thus, delayed activation of γδ T cells and MAIT cells and systemic activation of

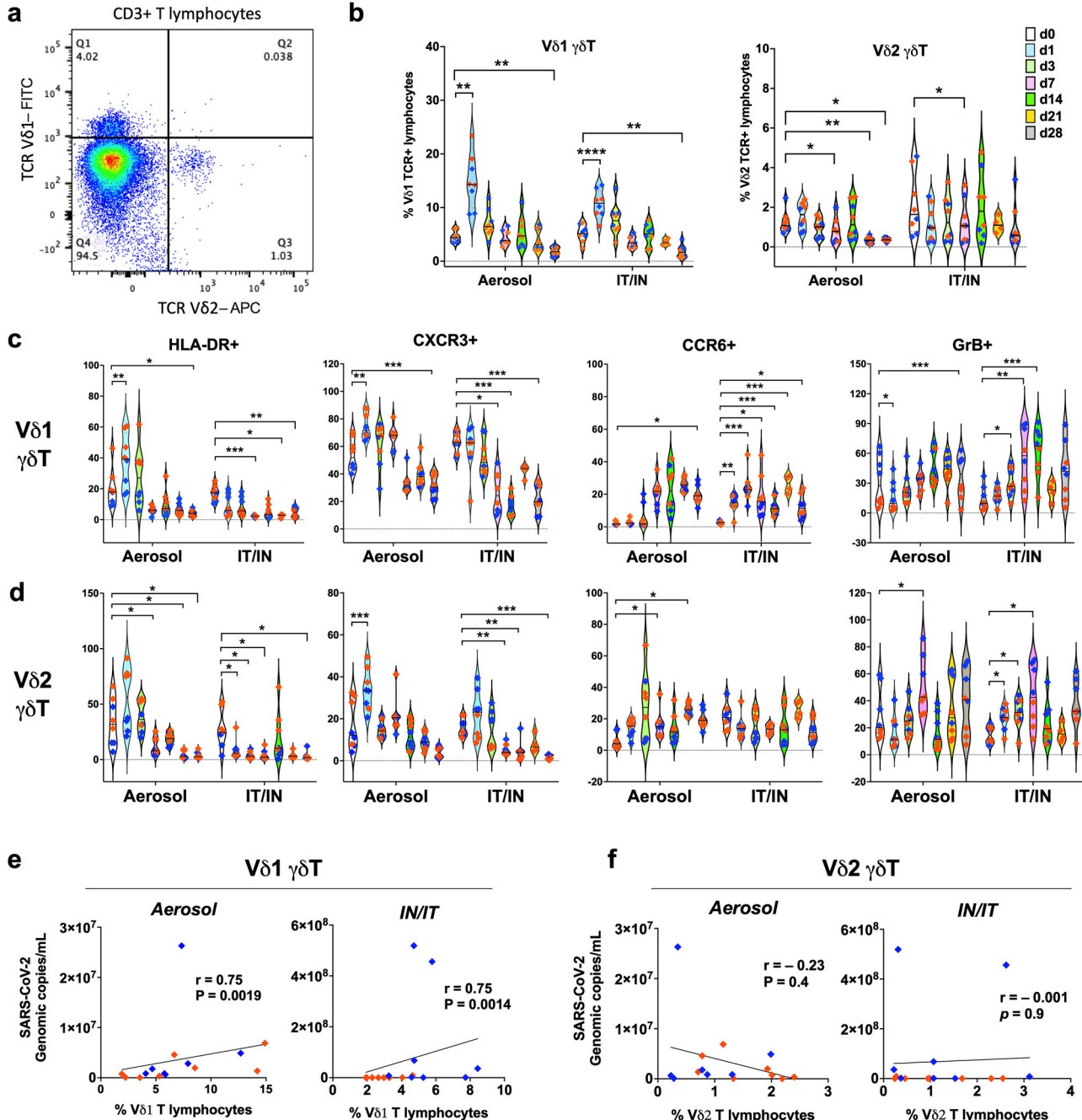

**Fig. 5 Dynamic changes in circulating Vδ1 and Vδ2 γδ T cell subsets and association with BAL viral loads during early SARS-CoV-2 infection.**
**a** Representative FACS plot of PBMC showing gating for Vδ1 and Vδ2 γδ T cell subsets in CD3 + T lymphocytes. **b** Frequencies of Vδ1 (left) and Vδ2 T cell subset (right) in PBMC following SARS-CoV-2 infection via aerosol or IT/IN routes (n = 8/group) on days 0, 1, 3, 7, 14, 21, and 28. Expression levels of HLA-DR, CXCR3, CCR6, and GrB on Vd1 T cells (**c**), and Vd2 T cells (**d**). Graphs show mean and SEM. Comparisons of different time points with respect to d0 baseline data were done using two-way ANOVA with mixed effects model and Dunnett's post hoc tests. Asterisks indicate significant differences between time points (*p < 0.05; **p < 0.01; ***p < 0.001; ****p < 0.0001). Spearman Rank correlation between BAL viral loads in the first week (d1 and d7) with corresponding frequencies of Vδ1 T cells (**e**), and Vδ2 T cells (**f**) showing significant correlations for Vδ1 T cells in both aerosol and IT/IN groups (n = 16/group). Individual values shown for each animal as symbols in orange for RMs and blue for AGMs.

CD8 T cells paralleled the early inflammatory response in lungs of IT/IN exposed animals.

Further evaluation of the correlations between γδ T cell frequencies and BAL cytokines in aerosol vs IT/IN exposure groups revealed that levels of MCP-1 (monocyte chemotactic factor regulating the migration and infiltration of monocytes, memory T cells, and NK cells) significantly correlated with BAL Vδ1 T cell frequencies at 14dpi in both routes of exposure

(Fig. 8b). However, significant negative correlations of Vδ1 T cell frequencies were noted with MIP-1α (Fig. 8b), also a monocyte/ macrophage chemotactic cytokine. Since MIP-1α is reportedly higher in mild to severe cases than asymptomatic SARS-CoV-2 infections[29], this suggests a positive association of lung Vδ1 T cell numbers with asymptomatic infection. Notably, aerosol infection displayed a significant correlation of Vδ1 T cells with IL-1Ra, which blocks inflammatory effects of IL-1 cytokines (Fig. 8b).

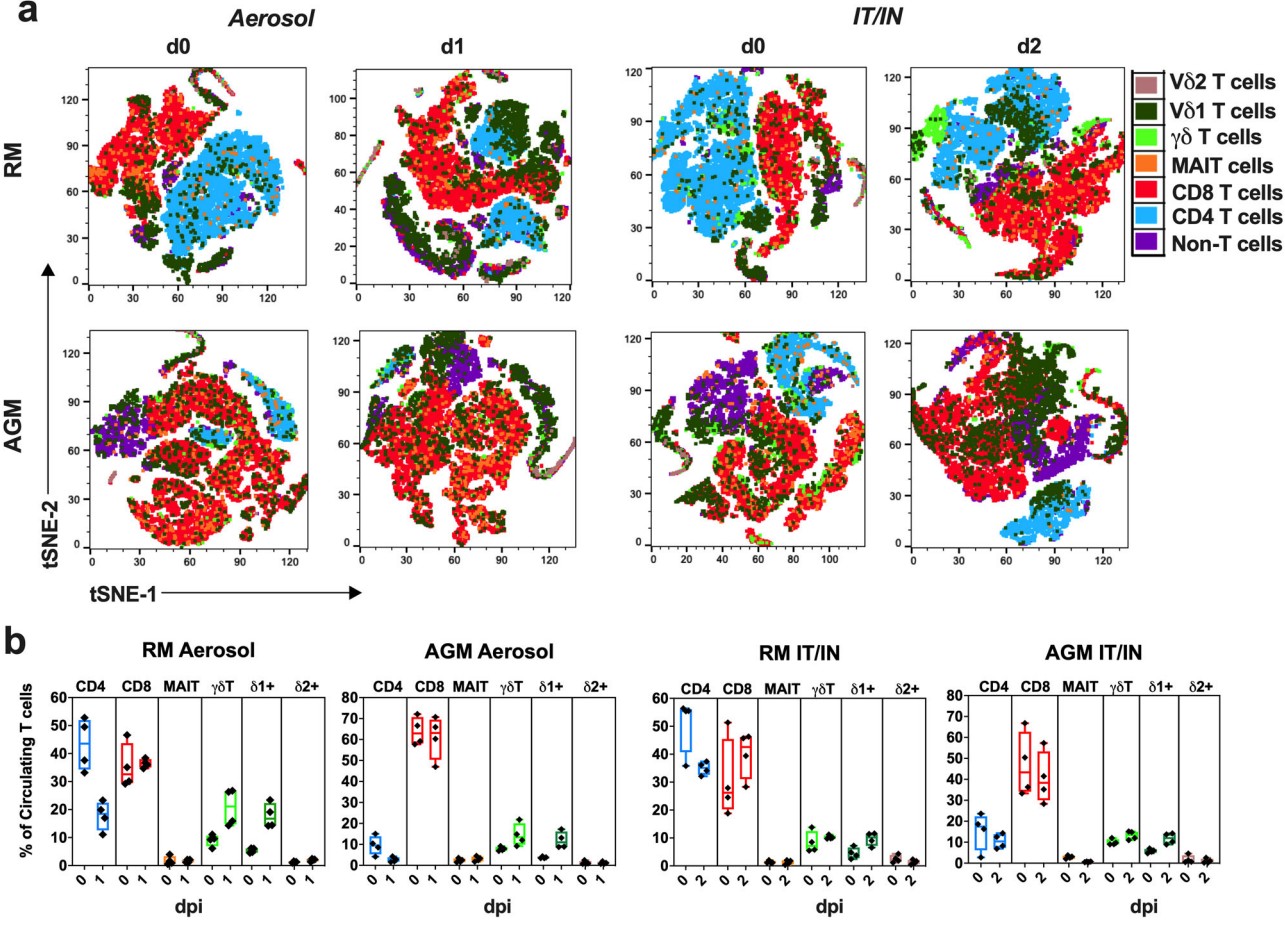

**Fig. 6 Atlas of T cell perturbations in NHPs immediately following SARS-CoV-2 infection via aerosol and IT/IN routes. a** t-distributed stochastic neighbor embedding (tSNE) plot of T cells based on flow cytometry data of CD3[+], CD4[+], CD8[+], γδ TCR[+], Vδ1 TCR[+], Vδ2 TCR[+], and CD161[+]/Vα7.2 TCR[+] cells in PBMC to visualize phenotypic changes occurring in circulating T cell populations immediately post-infection. T cell tSNE plots shown for d0 (left) and d1 (right) in the aerosol group and d0 (left) and d2 (right) in the IN/IT group. **b** T cell subpopulation frequencies for each tSNE plot showing comparison of day 0 versus 1 dpi in aerosol groups (n = 4/species) or day 0 versus 2 dpi in IT/IN groups (n = 4/species) of RM and AGM. Graphs show mean and SEM.

Interestingly, correlations of BAL Vδ1 T cells with CXCL13, a B cell activating chemokine that is also linked with severe COVID-19[40], and stem cell factor (SCF) were discordant between the aerosol and IT/IN routes, with only aerosol group displaying a significant positive correlation (Fig. 8b). No significant correlations of these cytokines/chemokines with other T cell subsets were observed in BAL fluid, suggesting that besides cytotoxic effector functions, Vδ1 T cells may also have a significant role in the local inflammatory response during asymptomatic/mild SARS-CoV-2 infection.

Overall, from these findings, a heuristic model of γδ T cell responses to SARS-CoV-2 infection emerges as shown in Fig. 9. The rapid activation (HLA-DR[+]) and expansion of circulating γδ T cells with increased CXCR3 expression in the aerosol group suggests their migration to naso-pharynx and lungs, the sites of high viral loads between 1–7 dpi (Supplementary Fig. 1). IT/IN exposure resulted in relatively delayed activation of blood γδ T cells at 3dpi, with increase in CCR6 expression suggesting an earlier induction of Th17-type functions in the setting of early development of clinical symptoms. During later stages, between 1-4 weeks of infection, a significant decline in blood γδ T cells and consistently increased GrB and CCR6 expression in both routes suggests induction of cytotoxic potential and increased homing to lungs during the recovery phase. Similarly, in the lungs, greater frequencies of HLA-DR[+], GrB[+] and CD161[+] γδ

T cells during the recovery phase may be involved in viral clearance and repair of alveolar epithelium. The late stage decline in γδ T cells likely represents AICD. The difference in routes of exposure and kinetics of γδ T cell mobilization may underlie the diverse clinical spectrum of COVID-19.

## Discussion

Given their diverse antimicrobial functions and enriched presence in mucosal tissues including lungs, γδ T cells have the potential to influence the quality and magnitude of early immune response to SARS-CoV-2 infection. Our study provides important insights into the potential role of Vδ1 T cells during early immune response to SARS-CoV-2 infection in NHP models of asymptomatic/mild disease. We demonstrate that γδ T cells are the first blood T cell subsets to activate and expand very early following aerosol SARS-CoV-2 exposure, with the Vδ1 subpopulation correlating with viral burden in the BAL fluid during active replication. Further, γδ T cells in BAL fluid display enhanced cytotoxic and Th17-type functional phenotype and significantly correlate with anti-inflammatory mediators in the lungs. Finally, we show that during the recovery phase of aerosol SARS-CoV-2 infection, Vδ1 T cells are significantly associated with immuno-modulatory factors IL1-Ra and MCP-1. While the increased cytotoxic responses are critical for control of viral replication, the shift toward epithelial barrier protective and immunoregulatory

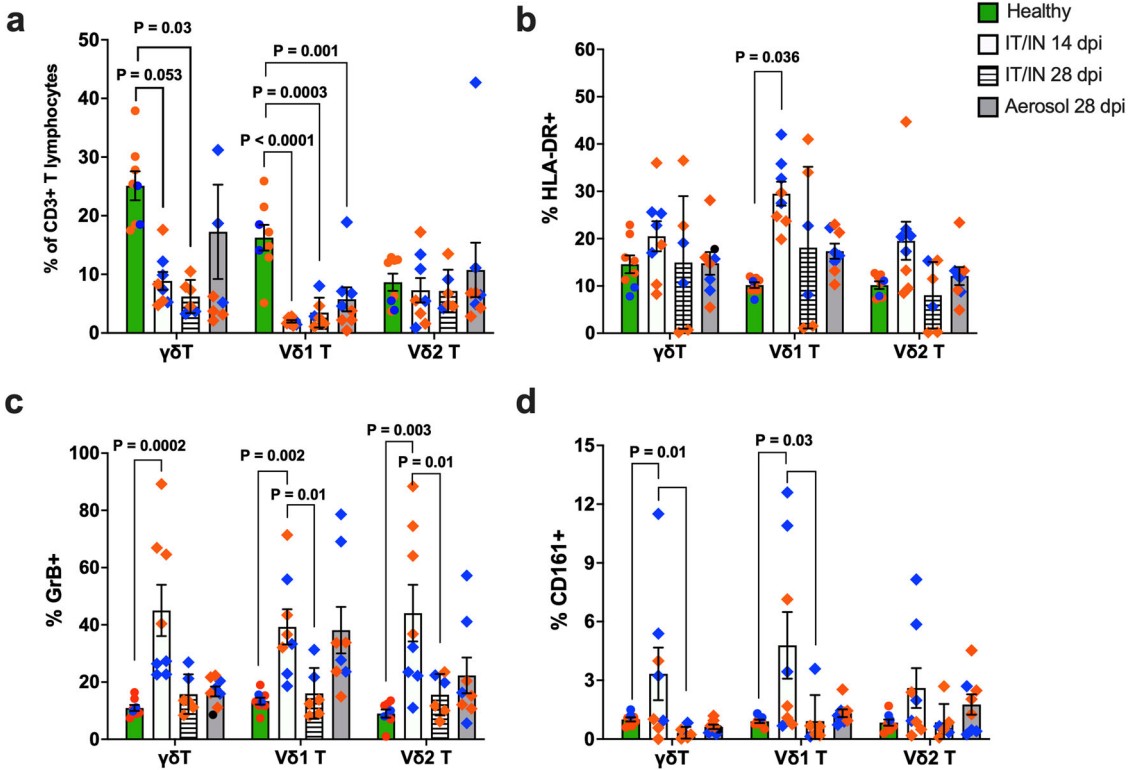

**Fig. 7 Frequency, activation and functional phenotype of γδ T cells in BAL fluid during SARS-CoV-2 infection. a** Cross-sectional comparison of frequencies of total γδ T cells, Vδ1, and Vδ2 T cells in BAL fluid from healthy nonhuman primates (*n* = 8/group) including 6 RMs (orange circles) and 2 AGMs (blue circles) with the study cohort of SARS-CoV-2 infected nonhuman primates (*n* = 8/group) comprising 4 RMs (orange diamonds) and 4 AGMs (blue diamonds) at 14 dpi and 28 dpi via IT/IN, or 28 dpi via aerosol challenge. Expression levels of HLA-DR (**b**), GrB (**c**), and CD161 (**d**) on total γδ T cells, Vδ1 T cells, and Vδ2 T cells in BAL (*n* = 8/group). Graphs show mean and SEM. Healthy and infected animals were compared using Mann Whitney test and longitudinal comparisons between 14 dpi and 28 dpi were done using Wilcoxon matched-pairs signed rank test.

functions could play an important role in preventing ARDS and alleviating clinical symptoms in patients with severe COVID-19.

Earlier studies have demonstrated a significant decline of blood Vδ2 T cells in severe and recovered COVID-19 patients[13,17,18]. The role of Vδ1 T cells, however, remains largely unknown. While the focus on Vδ2 subsets is not unwarranted considering their dominance in the peripheral blood of humans and well-documented antiviral functions against multiple viruses[10], Vδ1 subsets represent predominant tissue-associated γδ T cells with rapid effector functions[41]. Our results clearly demonstrate the rapid activation and expansion of circulating Vδ1 T cells following aerosol SARS-CoV-2 infection. This is further supported by a highly significant correlation of circulating Vδ1 T cell frequencies with BAL viral loads in the first week of infection in both aerosol and IN/IT routes of exposure. Indeed, Carissimo et al., have shown that during acute infection in COVID-19 patients, among T cell subpopulations including CD4, CD8 and γδ T cells, the highest levels of activated CD38+ effector memory and terminal effector cells are detected in Vδ1 T cells[13]. Expansion of Vδ1 T cells is also observed in other viral infections, including HIV/SIV[42–44], HBV[45], CMV[46], and human herpesvirus 8 (HHV-8) infections[47], likely due to in situ and systemic activation driven by infected cells that can produce cytokines like TNF-α, IL-1, IL-6, and IL-18, as well as IFN signaling during acute and chronic viral infections. The early expansion and increase in HLA-DR expression of Vδ1 T cells could be the result of a general increase in activated immune cells, although HLA-DR increase was first noted in Vδ1 T cells among T cell subsets. Notably, CMV-induced adaptive Vδ1 T cells have been recently shown to recognize HLA-DR on activated immune cells, pointing

to their role in inflammation-induced MHC-dependent immune response[48].

The later reduction at 28dpi in frequencies of overall γδ T cells, including Vδ1 and Vδ2 subsets, in both routes confirms that similar to severe COVID-19, the eventual decline of γδ T cells is a characteristic of asymptomatic/mild disease as well. It has been speculated that this decline in circulating γδ T cells denotes trafficking to lungs and other potential sites of viral replication. Intriguingly, however, our study revealed significant lower frequencies of total γδ T cells and Vδ1 T cells at 4 weeks in the BAL of SARS-CoV-2 infected NHPs. In the absence of BAL data during early infection in our study, it is unclear if there was an early increase in lung γδ T cells that subsequently declined owing to their relatively higher sensitivity to AICD[49–51] triggered by high levels of virus replication during the first two weeks of infection. The increase in many of the pro-inflammatory cytokines, including IL-6, TNF-α, GM-CSF, MIP-1β, MIG, IFN-α, and IL-1β in the BAL by 2 weeks and higher levels of activation and GrB+ Vδ1 T cells at 14dpi in comparison to 28dpi indicate an active role for Vδ1 T cells in local antiviral immune response via enhanced cytotoxic effector functions. During viral infections, γδ T cells can get activated by cytokines like TNF-α, IL-1, IL-6, and IL-18 produced by other immune cells[52] and upregulate the chemokine receptors such as CXCR3 and CCR5, enabling their recruitment to the site of inflammation, rich in CXCL9/10/11 and CCL3/4/5. The significant increase in CXCR3+ γδ T cells in our study at 1-7dpi in aerosol infection in contrast to IT/IN infection, indicated an impaired ability of Vδ1 T cells to traffic to sites of inflammation during early stages of IT/IN infection. Based on the immediate onset of clinical symptoms from 1 dpi of IT/IN

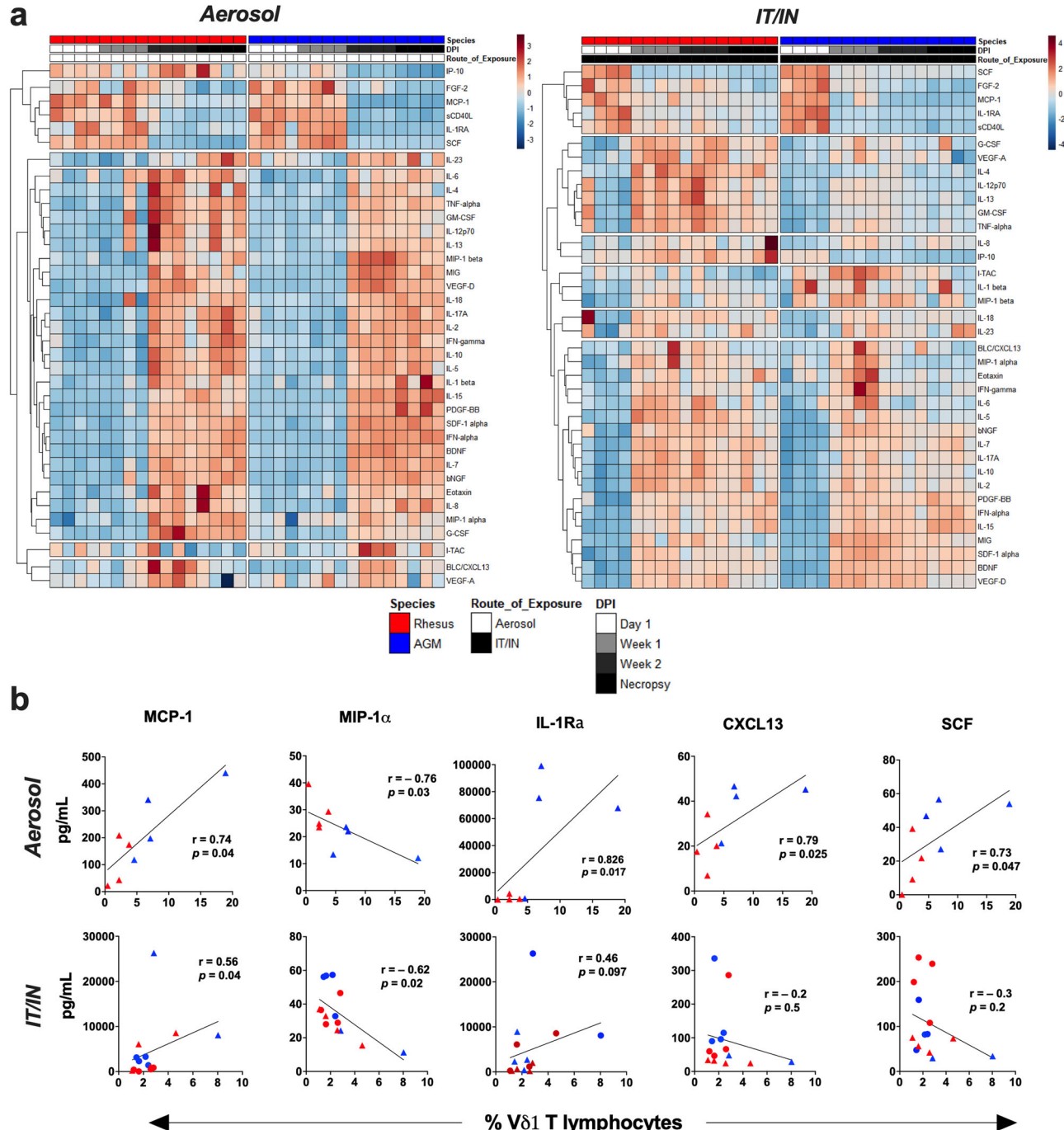

**Fig. 8 Cytokine responses in BAL fluid during SARS-CoV-2 infection and their correlation with local Vδ1 T cell frequencies. a** Lower airway cytokine and chemokine expression detected in bronchoalveolar lavage fluid following aerosol or IN/IT SARS-CoV-2 challenge represented by heatmaps. Each row of the heat map represents a different cytokine/chemokine, each column depicts an individual animal, and groupings of columns are used to organize study timepoints and species. **b** Spearman Rank correlation between BAL Vδ1 T cell subset frequencies with levels of MCP-1, MIP-1α, IL-1Ra, CXCL13, and SCF at 14 dpi and 28 dpi in aerosol group (n = 8/group; upper panel) and IT/IN group (n = 8–16/group; lower panel). Individual values shown for each animal as symbols in orange for RMs and blue for AGMs.

exposure in contrast to a delayed onset and milder clinical symptoms in the aerosol group in our study, we speculate that Vδ1 T cells were likely involved in the early control of SARS-CoV-2 pathogenesis following aerosol exposure.

Our results further suggest that the quality and kinetics of γδ T cell responses generated are influenced by the mode of SARS-CoV-2 exposure. Most notable was significantly higher circulating CCR6[+] γδ T cells in the IT/IN group starting early at 1-3dpi through 28dpi, in contrast to aerosol group that showed a delayed

increase in CCR6 around 7dpi. Since the IT/IN group displayed earlier and increased respiratory signs of disease, and a higher clinical score than the aerosol cohort[19], this suggests that direct droplet instillation on the mucosa, as opposed to disseminated viral exposure via aerosol, induced early localized signaling by airway epithelia and target cells to recruit Th17-type γδ T cells. CCR6 is an important marker for trafficking of epithelial barrier-protective IL-17A and IL-22-producing T cells, including γδ T cells to mucosal sites of inflammation such as gut, skin and

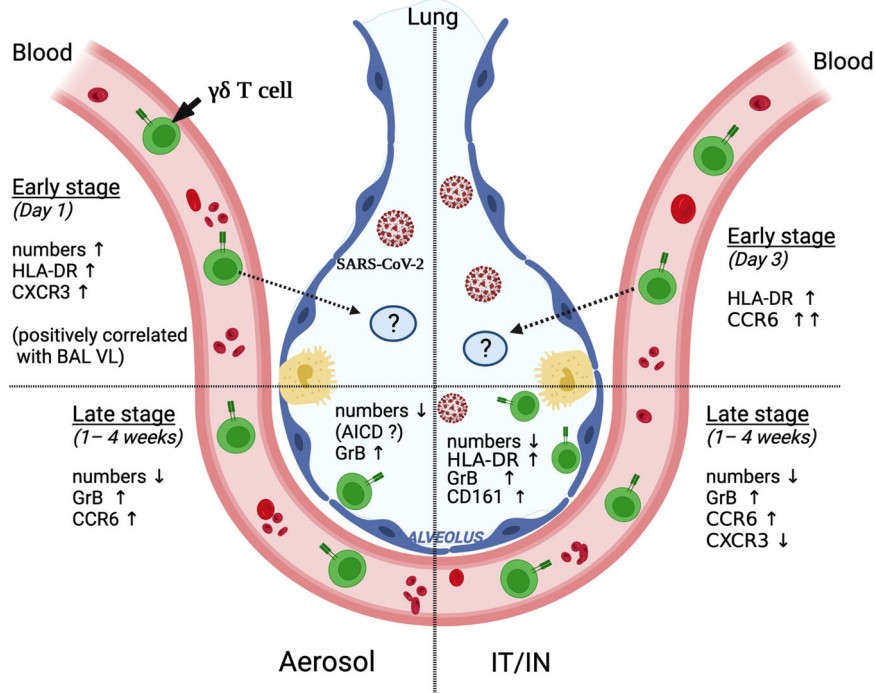

**Fig. 9 Peripheral blood and lung γδ T cell responses during SARS-CoV-2 infection.** Rapid activation (HLA-DR+) and expansion of circulating γδ T cells with increased CXCR3 expression, as observed following aerosol exposure in NHPs, may enable their migration to sites of virus replication in the airways. Later stages are marked by a significant decline in blood γδ T cells and consistently increased GrB and CCR6 expression in both routes of exposure, supporting killing of infected cells and repair of alveolar epithelia in the recovery phase. In the lungs, greater frequencies of activated (HLA-DR+), cytotoxic (GrB+) and CD161+ γδ T cells during the recovery phase may be involved in viral clearance and repair of alveolar epithelium. The differences in γδ T cell mobilization and effector functions may underlie the diverse clinical spectrum of COVID-19. *Schematic created with BioRender.com.*

lungs[30,53]. Along these lines, an earlier study in COVID-19 patients showed significantly greater levels of activated IL-17A-producing γδ T cells with decreased levels of IFN-γ-production among other innate T cells in the peripheral blood[17]. Thus, the early highly significant increase in CCR6+ γδ T cells in the IT/IN group likely represents the induction of an airway epithelia-protective immune response to clinical symptoms. CCR6+ γδ T cells have been implicated in lung tissue repair in influenza-infected neonates and in protecting the liver from excessive inflammation and fibrosis in the murine model of chronic liver injury[11,54]. Since IL-17 producing γδ T cells can also have pro-inflammatory functions[55], future studies are needed to elucidate the effects of CCR6+ γδ T cells on alveolar barrier integrity and SARS-CoV-2 responses using cell culture systems or explant culture model composed of epithelial, endothelial, and mononuclear cells.

A consistent feature between the two routes of exposure was the significantly increased GrB expression in Vδ1 T cells around 7dpi, suggesting that GrB cytotoxicity is a key anti-viral effector function of circulating γδ T cells during the first week of SARS-CoV-2 infection. Previously, Vδ2 T cells have been shown to inhibit in vitro SARS-CoV replication through an IFN-γ–dependent process[12]. Although direct killing of SARS-CoV-2-infected cells was not examined in our study, the significant increase in activated, GrB+ Vδ1 T cells recovered in BAL fluid at 14dpi and subsequent resolving of viral burden in the lungs and pharyngeal compartment suggests that Vδ1 T cells contributed to the decline in viral replication via GrB cytotoxicity. Our study further demonstrated that BAL Vδ1 T cells are significantly activated in contrast to blood at 14dpi, around the time of resolving viral loads in the lungs of most animals. Conversely, γδ T cells in BAL fluid and peripheral blood displayed similar enrichment of cytotoxic and Th17-type functional phenotype at

14dpi. This indicates that blood samples represented the overall functional changes in lung and airway γδ T cells, however, the levels of activation and later frequencies in BAL differed significantly from blood likely due to responses driven by local viral replication.

Although NHP models do not display clinical features of severe COVID-19 disease that may lead to ARDS[20–23], better understanding of T cell-responses in the context of resolved infections with mild disease is critical to delineate effective anti-viral immune responses in SARS-CoV-2 infected individuals. Of note, the early increase in activated, CXCR3+ γδ T cells in the aerosol exposure group with milder clinical symptoms and delayed BAL pro-inflammatory cytokines suggests that effective mobilization of γδ T cells may contribute to differences in tissue-specific immune response to SARS-CoV-2 regardless of peak viral loads. Further, our results showing increase in CCR6 between 3-7 days in both routes of inoculation that was sustained till up to a month and accompanied by decreased CXCR3 expression suggests a shift from Th1-type toward epithelial barrier protective functionality during the immune resolution phase of SARS-CoV-2 infection. This was supported by increased CD161 expression, another marker for Th17-type function, on BAL γδ T cells at 2 weeks besides a significant negative correlation with MIP-1α. Given that MIP-1α is found to be highly expressed in PBMC and BAL of COVID-19 patients[56], particularly with severe disease[57], and IL-1Ra has important regulatory effects on IL-1 mediated inflammatory responses, our results showing an inverse correlation of BAL Vδ1 T cell frequencies with levels of MIP-1α and positive correlation with IL-1Ra, suggest that Vδ1 T cells may be protective against developing severe disease by modulating early inflammatory signaling in response to SARS-CoV-2 infection in the lungs. Intriguingly, BAL Vδ1 T cell frequencies also correlated with levels of MCP-1, which is associated with inflammation and

pathogenesis of SARS-CoV-2[58,59]. In patients with mild COVID-19 symptoms, however, MCP-1 was shown to correlate with inhibition of IFN signaling via IRF3 downregulation[60]. Thus, in the context of NHP SARS-CoV-2 infection, this association concurs with a likely role of Vδ1 T cells in modulation of early inflammatory signaling. Additionally, the positive correlation with SCF, a key cytokine for the self-renewal and maintenance of haematopoietic stem cells (HSCs)[61] and prevention of lympho-penia, suggests that higher frequencies of Vδ1 T cells in the lungs during mild or asymptomatic infection is associated with better outcome to aerosol SARS-CoV-2 infection. Although these cor-relations point to the potential immunomodulatory role of Vδ1 T cells besides cytotoxic functionality during SARS-CoV-2 infection, future studies are required to establish a protective role against development of severe inflammation and ARDS of COVID-19. The recent identification of adaptive γδ T cells, particularly the Vδ1 γδ T cells, as long-term players in immunity has reshaped our view of γδ T cell responses to pathogens[62]. This is particularly relevant in long-COVID where most target cells in the lungs and other tissues, including brain and gut, might undergo major metabolic changes and inflammasome activation. Recent evidence points to an association of higher symptom load during acute phase of SARS-CoV-2 infection with long-COVID[63], and persistence of highly activated innate immune cells with elevated levels of type I IFN and type III IFN[64]. Our study provides a view into the dynamic changes in γδ T cell responses to early SARS-CoV-2 infection and suggests that rapid activation and migration of Vδ1 γδ T cells may contribute to protection from developing severe disease in the NHP model of aerosol SARS-CoV-2 exposure.

There are some limitations to this study, including the lack of data on baseline and early BAL γδ T cell responses within the first week of infection. Secondly, the low numbers of animals per exposure group per species diminished the statistical significance of comparisons between various groups within each species. Nevertheless, significant patterns consistent within each route of SARS-CoV-2 exposure emerged in two distinct NHP species combined, underscoring the conserved role of γδ T cells in the immune response to SARS-CoV-2 that can be directly applicable to humans.

In summary, our study shows that Vδ1 T cells mount a rapid, innate-like polyfunctional immune response to aerosol SARS-CoV-2 infection with early activation and mobilization to sites of inflammation and subsequent increase in Th17-type and cyto-toxic functionality. Overall, our study suggests the importance of future studies to determine the early functions of lung γδ T cells in mild versus severe COVID-19 cases to gain insights into their role in regulation of inflammation and virus clearance.

## Methods

**Ethical statement**. The Tulane University Institutional Animal Care and Use Committee approved all procedures used during this study. The Tulane National Primate Research Center (TNPRC) is accredited by the Association for the Assessment and Accreditation of Laboratory Animal Care (AAALAC no. 000594). The U.S. National Institutes of Health (NIH) Office of Laboratory Animal Welfare number for TNPRC is A3071-01. Tulane University Institutional Biosafety Com-mittee approved all procedures for work in, and removal of samples from, Biosafety Level 3 laboratories.

**Animals and infection**. A total of 16 adult (4-11 years old) male NHPs comprising of Rhesus macaques (*Macaca mulatta;* RM) and African green monkeys (*Chlor-ocebus aethiops;* AGM) were utilized for this study. RM were captive bred at TNPRC, and AGM were wild caught (St Kitts origin) and quarantined for at least 30 days prior to infection. Four individuals of each species were challenged with SARS-CoV-2 USA_WA1/2020 (World Reference Center for Emerging Viruses and Arboviruses, Galveston, TX), by small particle aerosol, with an average delivered dose of $1.5 \times 10^4$ TCID$_{50}$. The other four animals of each species were challenged via a combination of intratracheal and intranasal administration (IT/IN), with a dose of $2.0 \times 10^6$ TCID$_{50}$. Pre- and post-exposure samples were taken from blood,

mucosal (pharyngeal, nasal, rectal) swabs, and BAL supernatant. Animals were monitored for signs of disease throughout the study, with no animals reaching euthanasia criteria. At necropsy, mucosal samples were taken, as well as tissues placed in Trizol, Zinc-formaldehyde, or fresh frozen for later examination.

**Isolation of viral RNA**. RNA was isolated from non-tissue samples using a Zymo Quick RNA Viral Kit (#R1035, Zymo, USA) or Zymo Quick RNA Viral Kit (#D7003, Zymo, USA) for BAL cells, per manufacturer's instructions. RNA was eluted in RNAse free water. During isolation, the swab was placed into the spin column to elute the entire contents of the swab in each extraction. BAL supernatant was extracted using 100 μL. Viral RNA from tissues was extracted using a RNeasy Mini Kit (#74106, Qiagen, Germany) after homogenization in Trizol and phase separation with chloroform.

**Quantification of viral RNA using quantitative reverse transcriptase PCR**. Isolated RNA was analyzed in a QuantStudio 6 (Thermo Scientific, USA) using TaqPath master mix (Thermo Scientific, USA) and appropriate primers/probes (Supplementary Table 1) with the following program: 25 °C for 2 minutes, 50 °C for 15 minutes, 95 °C for 2 minutes followed by 40 cycles of 95 °C for 3 seconds and 60 °C for 30 seconds. Signals were compared to a standard curve generated using in vitro transcribed RNA of each sequence diluted from $10^8$ down to 10 copies. Positive controls consisted of SARS-CoV-2 infected VeroE6 cell lysate. Viral copies per swab were calculated by multiplying mean copies per well by amount in the total swab extract, while viral copies in tissue were calculated per microgram of RNA extracted from each tissue.

**Immunofluorescent staining and flow cytometry**. EDTA Blood was collected by venipuncture and bronchoalveolar lavage (BAL) was collected by bronchoscopy. The bronchoscope was introduced into the trachea and directed into each bronchus to instill two aliquots of 20 mL of saline and aspirate the fluids. Blood was layered over Ficoll-Plaque Plus (GE Healthcare17144003) and centrifuged for 30 min in cap-locked adapters to separate peripheral blood mononuclear cells (PBMCs) from other populations via differential migration of cells during cen-trifugation. PBMCs were collected, counted, and aliquoted in tubes or resuspended in freezing media containing Fetal Bovine Serum and DMSO and cryopreserved for later use. For BAL, fresh samples were filtered through a 100 μm strainer followed by separation of cells and supernatant by centrifugation (15 min at 2000× *g* at RT) in cap-locked adapters. RBCs were lysed in Ammonium Chloride Potassium (ACK) lysing buffer at room temperature for 5 min (Gibco# A1049201) and the remaining cells were washed in 2% FBS and resuspended in PBS and counted under laminar flux hood in BCL3.

Fresh or frozen cells were stained for flow cytometry with appropriate surface antibody cocktails and incubated for 30 min at room temperature. After subsequent washing with Running Buffer, cells were resuspended in Fixation/Permeabilization Buffer (BD Biosciences Cat# 554722) for one hour then washed with Perm/Wash Buffer (BD Bioscience Cat# BDB554723) twice. Intracellular target antibodies were then added and incubated for 20 minutes at room temperature, then washed with Running Buffer. Viability stain (BD Bioscience Cat# 564997) was used to differentiate live and dead cells. Commercial antibodies were used at manufacturer-recommended concentrations. Source of antibodies and clone information are listed in Supplementary Table 2. Samples were resuspended in FACS Fixation and Stabilization Buffer (BD Biosciences Cat# 50-620-051) and analyzed within 24 hours after sample processing. Unstained samples were run with every set of samples. Samples were run on the 3-laser, LSR Fortessa (BD Biosciences) and data were analyzed using FlowJo software (TreeStar).

**BAL cytokines and chemokines**. For evaluating cytokines and chemokines, Invitrogen 37-Plex NHP ProcartaPlex kits were purchased and processed according to manufacturer's instructions with a 1-hour sample incubation period and analysis on a Bio-Plex 200 software Bio-Plex Manager™ v6.2 (Bio-Rad, California, USA). BAL Samples underwent an additional overnight sample incubation period and fixation of the plate for one hour in 2% paraformaldehyde before resuspension in Reading Buffer and analysis on Bio-Plex 200 software Bio-Plex Manager™ v6.2. Heatmaps were generated using log2-transformed raw fluorescent intensity values input into the R package pheatmap (Raivo Kolde 2019. pheatmap: Pretty Heat-maps. R package version 1.0.12.) with unsupervised Hierarchical clustering analysis.

**Statistics and reproducibility**. Statistical analyses were performed using Prism 9.1.2 (Graphpad Software, LLC.) Within-group comparisons for immune responses at different time points with respect to baseline data were performed using repeated measures one-way ANOVA with Dunnett's post hoc tests. Cross-sectional com-parison of frequencies and phenotype of γδ T cells in BAL were performed using Mann Whitney test. All correlations were computed using nonparametric Spear-man rank correlation test. P values of 0.05 or lower were considered significant, ∗$p < 0.05$, ∗∗$p < 0.01$, ∗∗∗$p < 0.001$, ∗∗∗$p < 0.0001$. Polyfunctional responses were compared using SPICE 6 software[65]. Cytokines and chemokines measurements included at least two technical replicates.

**Reporting summary**. Further information on research design is available in the Nature Portfolio Reporting Summary linked to this article.

## Data availability

All data supporting the findings of this study are within the article and its Supplementary Information files or are available from the authors upon request. The raw data supporting the findings are provided as Supplementary data.

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

## Acknowledgements

We are grateful to all the veterinary personnel, administrators, and laboratory managers who were involved in this work. We thank Drs. Scott Weaver, Kenneth, and Jessica Plante at University of Texas Medical Branch for provision of the SARS-CoV-2 viral stock used for the innoculations in this study. We acknowledge Mary Barnes and Melissa Pattison of the Pathogen Detection and Quantification Core of Tulane National Primate Research Center (TNPRC) for assistance with the multiplex cytokine detection assays and use of the Bioplex-200 instrumentation, and the clinical veterinary staff in the Division of Veterinary Medicine at TNPRC for coordinating the biospecimen collections. Technical assistance of the flow cytometry core facility staff at the TNPRC is greatly appreciated. We thank Angela Birnbaum in the Office of Biosafety for reviewing and optimizing all technical SOPs and overseeing the safety of this study. We are extremely grateful to the funding from NIAID Contract HHSN272201700033I; NIH ORIP Grant P51OD011104; NIAID grants R21AI140840, and R33AI136100; and NIGMS grant P20GM103629.

## Author contributions

Conceptualization: N.R., C.J.R. Methodology: A.C.F., E.W., N.C., N.S., N.G., B.P., S.S., K.E.R.-L., L.A.D.-M., R.V.B., B.J.B., N.J.M. Investigation: N.R., A.F., K.M.M., S.S., R.V.B. Visualization: N.R., A.F., K.M.M. Funding acquisition: C.J.R., N.R. Project administration: C.J.R., N.R. Supervision: C.J.R., N.R., L.A.D.-M. Writing – original draft: N.R. Writing – review & editing: N.R., C.J.R. All authors reviewed and approved the final manuscript.

## Competing interests

The authors declare no competing interests.
