## [Peer Review File · Communications Biology]

Reviewers' comments:

Reviewer #1 (Remarks to the Author):

Comments to authors:

Here the authors have very thoroughly studied the phenotypes and functionalities of gd T cells in peripheral blood and BAL in SARSCov2 infected African green monkeys and rhesus macaques. The authors find increased frequencies of particular gd T cell subsets and the authors argue this may contribute to the antiviral immune response. Overall the studies are well performed, appropriate controls are included, and the conclusions are justified by the data presented.

1. Saying gd T cells are "innate" is a little misleading. They do have rearranged TCRs. This should be rephrased.
2. There are numerous papers showing that VD1s are expanded in SIV and HIV infections. The authors should reference and discuss these previous findings as they might point to a potential mechanism underlying the phenomenon. Maybe IFN signalling, or some other aspect of inflammation that is shared between viral infections? Are there any data in other viral infections (aside from SARSCov2 and HIV/SIV)?
3. There are new data suggesting gd T cells can recognize HLA-DR. This work should be cited and discussed as it may explain the authors' findings.
4. The correlations in Figure 3C are almost certainly attributed to AGM having lower frequencies of CD4 T cells in general (due to down-regulation of CD4 by memory CD4 T cells). Thus, that correlation should be omitted, and the corresponding discussion removed from the manuscript.
5. In Figure 4 it would be helpful to know if other T cells also have differential expression of CXCR3 and CCR6. The authors should gate on the ab T cells (at least TCR gd negative) and show frequencies of CXCR3 and CCR6.
6. The authors should make clear in the figure legends which analyses are from PBMC and which are from BAL.

Reviewer #2 (Remarks to the Author):

This is a very well written report. It builds on good immunological analyses of the role of gdT cells in different stages of non-human primates. This is very important as rodents have different subset uses to ours and non-human primates. The conclusions are well underbuilt by results and are well described in text as well as figures. Conclusions are novel and convincing and point to the clear involvement of the d1 subset.

It could have been nice to see some data on the cytotoxic direct functionality and pAPC function that are alluded to in the report. As it stands it is somewhat limited in that respect, albeit very concise and thorough in terms of phenotypic analysis.

Minor points:

Line 298: "the shift toward epithelial barrier protective and immunoregulatory functions" - these functions are interesting and should be discussed further in terms of implications.

Line 350 and elsewhere (including introduction): it is unclear what exactly the different exposure routes mean in terms of viral uptake, initial exposure to immune system etc. These aspects should be discussed as they are fundamental to the comparisons made.

Response to reviewers' comments:

Reviewer #1 (Remarks to the Author):

We thank the reviewer for the detailed comments to improve this manuscript.

Comments:

1. *Saying gd T cells are “innate” is a little misleading. They do have rearranged TCRs. This should be rephrased.*

We agree and thank the reviewer for this comment. We have now rephrased it to “innate-like” T cells, since gdT cells do have a shared innate and adaptive function, and the kinetics of the response observed in our study is very early. So, we believe that the initial response to SARS-CoV-2 at day 1 is very likely an innate-like response to stress-induced ligands on infected cells and the later response may be more adaptive in nature, an aspect that we intend to explore in future studies of re-infection and expansion of specific TCR clonotypes.

2. *There are numerous papers showing that VD1s are expanded in SIV and HIV infections. The authors should reference and discuss these previous findings as they might point to a potential mechanism underlying the phenomenon. Maybe IFN signalling, or some other aspect of inflammation that is shared between viral infections? Are there any data in other viral infections (aside from SARSCov2 and HIV/SIV)?*

Indeed. We were motivated from our and previous studies of SIV/HIV infections to explore Vdelta1 subsets in this study. We have now discussed these findings in the broad context of viral infections of mucosal tissues and cited relevant references (lines: 312-317). We thank the reviewer for enriching the discussion!

3. *There are new data suggesting gd T cells can recognize HLA-DR. This work should be cited and discussed as it may explain the authors' findings.*

Thank you for pointing this out. We have now discussed this in lines 317-321 and cited relevant reference.

4. *The correlations in Figure 3C are almost certainly attributed to AGM having lower frequencies of CD4 T cells in general (due to down-regulation of CD4 by memory CD4 T cells). Thus, that correlation should be omitted, and the corresponding discussion removed from the manuscript.*

We agree that the negative correlation in Fig. 3C is influenced by AGM having lower frequencies of CD4 T cells. Based on the suggestion, we have now removed 3C and related discussion.

5. *In Figure 4 it would be helpful to know if other T cells also have differential expression of CXCR3 and CCR6. The authors should gate on the ab T cells (at least TCR gd negative) and show frequencies of CXCR3 and CCR6.*

We have now included the data on CXCR3 and CCR6 expression by CD4 T, CD8 T, and MAIT cells as Supplementary figure 3. No consistent differential expression patterns were observed in the ab T cells (TCRgd-negative cells) including MAIT cells.

6. *The authors should make clear in the figure legends which analyses are from PBMC and which are from BAL.*

We have now clarified which analyses are from PBMC or BAL in the figure legends.

Reviewer #2 (Remarks to the Author):

We thank the reviewer for the enthusiastic support for this work and the kind words of appreciation!

Comments:

1. *It could have been nice to see some data on the cytotoxic direct functionality and pAPC function that are alluded to in the report. As it stands it is somewhat limited in that respect, albeit very concise and thorough in terms of phenotypic analysis.*

We strongly agree with the reviewer's comment. This is a very important experiment that we plan to conduct in future studies. Unfortunately, the cryopreserved cells from this study were lost due to extended power outage during hurricane Ida, making any further data acquisition to address this comment with samples from the present study, a mechanical impossibility.

2. *Line 298: "the shift toward epithelial barrier protective and immunoregulatory functions" - these functions are interesting and should be discussed further in terms of implications.*

We completely agree that this is the most interesting finding of this study and based on the reviewer's suggestion, we have now discussed this further in addition to what we had already discussed as implications for airway inflammation and ARDS of COVID-19 (lines 365-388 in the previous draft: highlighted now in lines 379-408).

3. *Line 350 and elsewhere (including introduction): it is unclear what exactly the different exposure routes mean in terms of viral uptake, initial exposure to immune system etc. These aspects should be discussed as they are fundamental to the comparisons made.*

This is a great comment. A recent publication (published in the interim) from this work details the rationale and COVID-19 disease characteristics in the two NHP species challenged with SARS-CoV-2 by direct mucosal (intratracheal + intranasal) instillation or small particle aerosol in route-discrete sub-cohorts within each species (Fears et al., PLoS Pathog. 2022 Jul 5;18(7):e1010618. doi: 10.1371/journal.ppat.1010618.) This publication demonstrated the differential outcome (pleurisy) based upon modality suggesting that the viral uptake and mode of initial exposure to the immune system may indeed impact the clinical outcome. The current manuscript under consideration is a more focused investigation of the very early T cell immune response in blood and BAL fluid to understand conserved immune responses that may contribute to reduced pulmonary pathology. We have now added text (lines: 347-351) discussing this and referenced the related publication (replaced the biorxiv citation with the publication).

REVIEWERS' COMMENTS:

Reviewer #1 (Remarks to the Author):

The authors have addressed the concerns raised by both reviewers.